# Performance Efficiency of Conventional Treatment Plants and Constructed Wetlands towards Reduction of Antibiotic Resistance

**DOI:** 10.3390/antibiotics11010114

**Published:** 2022-01-16

**Authors:** Moushumi Hazra, Lisa M. Durso

**Affiliations:** 1Department of Hydrology, Indian Institute of Technology, Roorkee 247667, Uttarakhand, India; 2Agroecosystem Management Research Unit, Agricultural Research Service, United States Department of Agriculture, Lincoln, NE 68583, USA; lisa.durso@ars.usda.gov

**Keywords:** conventional treatment plants, wastewater, treated effluent, constructed wetlands, antibiotic-resistant genes/bacteria, macrophytes

## Abstract

Domestic and industrial wastewater discharges harbor rich bacterial communities, including both pathogenic and commensal organisms that are antibiotic-resistant (AR). AR pathogens pose a potential threat to human and animal health. In wastewater treatment plants (WWTP), bacteria encounter environments suitable for horizontal gene transfer, providing an opportunity for bacterial cells to acquire new antibiotic-resistant genes. With many entry points to environmental components, especially water and soil, WWTPs are considered a critical control point for antibiotic resistance. The primary and secondary units of conventional WWTPs are not designed for the reduction of resistant microbes. Constructed wetlands (CWs) are viable wastewater treatment options with the potential for mitigating AR bacteria, their genes, pathogens, and general pollutants. Encouraging performance for the removal of AR (2–4 logs) has highlighted the applicability of CW on fields. Their low cost of construction, operation and maintenance makes them well suited for applications across the globe, especially in developing and low-income countries. The present review highlights a better understanding of the performance efficiency of conventional treatment plants and CWs for the elimination/reduction of AR from wastewater. They are viable alternatives that can be used for secondary/tertiary treatment or effluent polishing in combination with WWTP or in a decentralized manner.

## 1. Introduction

Wastewater treatment plants (WWTPs) are a central distribution point where effluents are distributed to receiving water bodies. WWTPs are efficient for the reduction of many pollutants but have not shown efficiency for the removal of antibiotic-resistant bacteria (ARB) and antibiotic-resistant genes (ARGs) [1,2]. Municipal wastewater contains a rich microbial community derived from feces and other sources, including both naturally occurring and anthropogenically enriched ARBs, along with unmetabolized drugs and their metabolites which are secreted in urine and feces [3,4]. The presence of heavy metals, antibiotic residues, and other emerging contaminants add to the load in WWTP that facilitates conditions for horizontal gene transfer and accelerates the development of ARB, including uptake of ARGs by pathogens. WWTP effluents and biosolids are directly discharged to water bodies, agricultural lands, or soil, and there is growing evidence that treated effluents transport ARBs and ARGs through surface runoff, discharge to water bodies and percolating into groundwater [5,6].

The United Nations Environment Programme identifies antibiotic resistance among the top emerging issues of concern [7], and the human health and economic impacts have been well documented [8]. The development of antibiotic resistance is a natural evolutionary process mediated by microorganisms and accelerated by selective pressures due to anthropogenic activities [9]. For example, the soil is a natural reservoir of ARBs and ARGs with potential impacts on human health [10], but ARB and ARG targets are enriched in soils following the application of biosolids or animal manures [11,12]. ARGs have been detected in many environmental compartments, including in river water [13], sediment [14], soil [15,16], and both the Antarctic and Arctic ecosystem [17,18,19]. Hotspots for the enrichment of antibiotic resistance include not only medical settings but also environmental compartments that are subjected to anthropogenic pressure [20], such as urban and municipal wastewater systems, pharmaceutical manufacturing effluents [21], aquaculture facilities and animal husbandry facilities [22]. Like municipal wastewater, rural and agricultural wastewater is also characterized by high bacterial loads coupled with excreted antibiotics, and it contributes to the discharge of ARB and ARGs into the environment [23]. One key challenge in regards to municipal, industrial and agricultural wastewaters is that they have many entry points into the environment. Figure 1 highlights the source, development, pathway and potential threat of antibiotic resistance on the environment and application of constructed wetlands for reduction of the same.

Constructed wetlands (CWs) are a sustainable “green” wastewater treatment option that can be used as an alternative to WWTP or as a component of conventional WWTP. They use natural processes, including substrates, plants, soils, and microbes, to filter and treat pollutants in water. Recent studies on CWs have gained significance due to their ability to treat pharmaceuticals [24,25] from municipal and hospital wastewater [26,27]. There is evidence to suggest that CWs can be an effective mitigation strategy for antibiotic drugs [28], ARB, and ARGs [29,30], as well as organic pollutants and heavy metals [31,32] that have also been postulated to select for antibiotic resistance [33]. CWs can be used alone or even after primary treatment, and they have been reported to show a 2 log [34] to 4 log reduction [35] of pathogens with various configurations and designs resulting in different removal efficiencies. When coupled as secondary/tertiary units to primary treatment in WWTP, CWs assist in enhancing the performance of the WWTP [36]. On the basis of path flow/hydrology, CWs are classified as vertical flow (VFCW), horizontal subsurface flow (HSSFCW), surface flow (SFCW), and hybrid systems, generally known as integrated flow CW (IFCW) [37]. Appendix A represents the various types of CWs according to design, specification and types of plant species used.

There is growing interest in the functionality of CWs for the reduction of antibiotic drugs [38,39], ARBs [40], and ARGs [41]. Recent studies have focused on the use of CWs for the polishing of effluents from conventional treatment plants, aiming to achieve the further reduction of antibiotic residues [42] and coliforms [43,44,45]. The construction, operation and maintenance costs are significantly lower when compared to conventional WWTP options [46], making CW an economical and practical option, especially for developing/low-income countries [47]. Another added advantage is that CWs can be used as a decentralized option both in rural and urban areas, providing aesthetically pleasing sites that promote biodiversity and provide wildlife habitats with minimum daily upkeep requirements [48]. As such, CWs contribute to multiple UN sustainable development goals, including clean water and sanitation, good health and well-being, industry innovation and infrastructure, sustainable cities and communities, life on land, and life below water [49].

Given the urgent need to sustainably address antibiotic resistance, the potential for wastewater treatments as an antibiotic resistance critical control point, and encouraging data gathered to date on the potential of CWs for mitigating antibiotic drugs, ARB and ARGs in the waste stream, the present study was conducted with the specific objective to review and explore the performance efficiency of CWs towards multiple measures of antibiotic resistance, identify the major advantages/benefits CWs provide to the environment and compare CWs with conventional treatment plants for the reduction of antibiotic-resistant targets and other microbial pollutants. The present study will therefore provide a backbone for better understanding and evaluating the challenges faced by developing/low-income countries when managing and disposing of their huge amounts of wastewater, with a focus on antibiotic resistance outcomes.

## 2. Methods

### 2.1. General Approach

In order to understand the major problems associated with the disposal of wastewater in the context of antibiotic resistance in developing/low-income countries and to identify knowledge and research gaps, a literature search was conducted. We considered recently published articles for the abundance of ARGs in different environments between 2016–2021 and also included some significant studies reported during 1981–2015. The keywords used were CW AND ARGs AND bacteria AND WWTP AND mechanism of ARB in CWs AND AR in sludge AND AR in WWTP. Initially, articles and reviews based on effluents discharged from treatment plants (with/without tertiary units and chlorination), their characteristics and their impact on the environment were identified. We were also focused on antibiotic resistance associated with WWTPs, sludge being used as fertilizers without any post-treatment, CWs, treated effluents (both from treatment plants and CWs) and their effect on the receiving water bodies. There are no discharge standards for AR for effluents that are being directly discharged to water bodies and agricultural land for irrigation. Moreover, there is ignorance regarding the treatment of sludge used as fertilizer or disposed to nearby land. Then the question of concern/interest was framed in a detailed manner, reviewing the significance of each point dealt with in the study and its significance (Table 1). The performance and the mechanism taking place in CWs were studied in detail through the reconnaissance of literature as described below.

### 2.2. Bibliography Approach and Analysis

Previous literature was studied with a focus on papers describing measures of antibiotic resistance (including ARBs and ARGs) in CWs, WWTPs, wastewater, treated effluent, sludge, anaerobic digestion and activated sludge units of treatment plants, cattle feedlots and slaughterhouses [63]. A brief schematic in Appendix A demonstrates the basic steps conducted for the present review.

## 3. Review Findings

The present study is a literature review of studies related to antibiotic resistance in CW and WWTPs and its effluent discharge to water compartment, which is a major challenge in developing/less developed countries. Journal articles were identified in Scopus (*n* = 65), Web of Science (*n* = 55), Science Direct (*n* = 33), Google Scholar (*n* = 15) and also scanned references (*n* = 31). The total number of journals articles screened for the present review was 199. This included publications related to WWTPs (74), CWs (59), rural and urban wastewater (28), hospital wastewater (12), and aquatic and soil ecosystems (26). We then framed the question of concern regarding the protocols for antibiotic resistance, restrictive use of antibiotics, treatment of wastewater, effluent discharges and use and implementation of natural treatment system for ARG/ARB removal with a focus on ARGs in wastewater. Hotspots of antibiotic resistance in the environment and their pathway of exposure/disposal were identified. The recent studies were subdivided into (i) clinical studies based on case studies reported in humans, plants and animals, and (ii) environmental studies based on lab and field studies on water, soil, agricultural farms, poultry and anthropogenic activities. Then, information pertaining to the performance efficiency of WWTPs and CWs for the elimination of antibiotic resistance was consolidated. There is an increasingly robust body of literature on CWs and their potential technology that assists in the removal of antibiotic resistance. Studies reported during 2019 and 2021 demonstrate encouraging use of CWs for the reduction of general water and microbial pollutants. It is expected that the application of CWs may further increase in the near future due to its multiple benefits such as aesthetics, ability to reduce a variety of pollutants, and capacity for different configurations to suit stakeholders’ requirements.

### 3.1. Main Drivers of Antibiotic Resistance in Wastewater

Antibiotic drugs are considered the main driver of antibiotic resistance, and they enter the wastewater stream via human, animal, medical, and industrial waste, along with heavy metals of different concentrations according to their sources. The sublethal concentration of antibiotics/heavy metals/biocides may alter/have a direct impact on microbial cell function as well as resistance properties [64]. These waste streams also contain enteric pathogens, coliforms, phages, ARB, and ARG, which then are combined during centralized waste treatment. All are routinely isolated from WWTP [65,66], which are considered hotspots of horizontal gene transfer of ARGs [9]. CWs are an alternative system for treating these waste streams, and there is a growing body of work exploring the occurrence, persistence, and reduction of antibiotic drugs, ARB and ARG in CW. There is a suite of drug-resistance types and bacteria that are a top priority for human health and environmental surveillance of antibiotic resistance. For example, *Enterococci* species are associated with feces and species such as *Enterococcus faecium* and *Enterococcus faecalis* have been linked to illnesses such as urinary tract infections and endocarditis [67]. *Enterococci* possess an intrinsic ability to resist a wide spectrum of antibiotics, including cephalosporins and aminoglycosides [68]. They also have a natural tendency to acquire and disseminate antibiotic resistance against a variety of antibiotics [69]. *Escherichia coli* that cause infections are generally more resistant than commonly found *E. coli* associated with fecal matter. Hence it is only possible to predict clinical resistance from wastewater analysis if the difference in resistance level between both strains (*E. coli* strains that cause infections and strains that are associated with fecal contamination) causing infections is more or less the same [70].

### 3.2. Efficiency of Conventional Treatment Plants for Reduction of Antibiotics, ARBs and ARGs

WWTPs are generally comprised of preliminary and primary treatment (screening and sedimentation basin) for the removal of suspended solids. The secondary treatment includes an activated sludge process and a biological contactor/filter process responsible for degrading/breaking organic compounds assisted by microbial metabolism. The tertiary treatment includes disinfection, CWs, sand filtration, a membrane bioreactor, etc. The treatment technology and use of tertiary treatment methods impact the prevalence of antibiotics, ARBs and ARGs.

#### 3.2.1. WWTPs and Antibiotic Reduction

Antibiotics are chemical pollutants and include drugs such as β-lactams, cephalosporins, fluoroquinolones, tetracyclines and sulfonamides, among others. They find their path to the WWTP from domestic, industrial, clinical, pharmaceutical and other effluent discharges [64,71,72]. Each type of antibiotic drug possesses different physicochemical properties that impact its adsorption and biodegradation process during wastewater treatment. In conventional WWTP, hydrophobic antibiotics like chloramphenicol and aminoglycosides [73] will get adsorbed onto the sludge, whereas hydrophilic antibiotics such as fluoroquinolones will be removed through electrostatic interactions with the cell membrane of microorganisms [74]. The authors of [75] investigated the incomplete removal of fluoroquinolones by WWTPs comprised of a conventional activated sludge process and membrane bioreactor. Maximum removal was observed in the range of 2–31%. In contrast, sulfamethoxazole was removed in the range of 4–56%, with maximum removal reported in a membrane bioreactor. Sulfonamides are insoluble in water. Less absorptive sulfonamides are mostly degraded in the oxic tank [76]. They are generally removed from the water phase by biodegradation and photodegradation. Trimethoprim is primarily removed by sediment adsorption [77]. The quinolone molecule is also strongly adsorptive and is soluble in water in the same manner as aminoglycosides. WWTPs have been shown to be effective in the removal of tetracyclines due to their hydrophilic nature [78]. Most tetracyclines that enter the system get adsorbed by activated sludge flocs, concentrated, and removed mainly by the secondary clarifier. The presence of certain nitrifying and oxidizing bacteria (Actinobacteria, Proteobacteria, Nitrospirae) in the activated sludge component of WWTP assists in the biodegradation of antibiotics [4], including *Dechloromonas* and *Thauera*, that have been identified as typical denitrifying bacteria in wastewater treatment [79]. Various studies have reported the removal of antibiotics with an advanced oxidation process [80,81]. Reverse osmosis and activated carbon (used for tertiary treatment) do not degrade or remove but transfer the antibiotics from one phase to another. However, the use of TiO_2_, ZnO, and CdS for photocatalysis has shown encouraging results for antibiotic removal due to their availability, nontoxic nature, low cost, strong oxidizing capability and use either as colloids or in an immobilized form [82].

##### Primary Treatment Data/Literature

Various works of literature have reported that primary treatment is ineffective for the removal of antibiotics with a low reduction of antibiotic residues [83]. Complexities exist in WWTPs where biodegradation, hydrolysis, and adsorption to sludge are the significant processes that assist in the elimination of antibiotics.

##### Secondary Treatment Data/Literature

Investigations report that the removal efficiency of a particular antibiotic varies significantly even within the same type of process, such as activated sludge or anoxic/oxic (A/O) conditions. The use of only activated sludge may not show better results as compared to a combination of activated sludge with anoxic and oxic conditions. Therefore, a hybrid system employing both enhances the removal of antibiotics such as lincomycin in the range of 58–74% [84]. It is a challenge to detect trace levels of antibiotics in the wastewater of WWTPs, where several factors such as pH, matrix interferences and other environmental factors (solar irradiation, temperature and precipitation) result in inaccurate detection. In a study reported by [85], the removal of antibiotics was investigated, considering the various operational parameters. They found that the removal of clindamycin and ciprofloxacin has a significant correlation with temperature and sludge retention time. Reduction of fluoroquinolones has been reported to be reduced >80% in WWTPs that employ activated sludge treatment methods [86], where the activated sludge treatment involves the biodegradation of antibiotics [87].

##### Tertiary Treatment Data/Literature

Tertiary treatment involves disinfection methods that assist in the removal of microbial contaminants and often involve chlorination. They are also found to be efficient for the removal of tetracycline, trimethoprim, erythromycin, ciprofloxacin, sulfamethoxazole and norfloxacin. As reported by [88], when comparing tertiary with secondary treatment, the antibiotic concentration is significantly reduced in the former with approximately 3% for tetracycline, 5% for norfloxacin, 11% for both sulfamethoxazole and ciprofloxacin, 17% for erythromycin and 23% for trimethoprim. Removal of approximately 93% [89] to a maximum of 99% [90] of trimethoprim has been reported, indicating good efficiency of tertiary units employing chlorination alone for the elimination of this particular antibiotic. A recent study has also reported the elimination of tetracycline, sulfonamides and macrolides by chlorination [91]. UV treatment, however, has not been considered efficient for the removal of antibiotics. About 15% removal of antibiotics is observed in UV, as compared to 45% removal in chlorination treatment as reported by [92]. Other treatment processes such as ozonation are quite expensive but effective at removing antibiotics. The authors of [93] have reported removal of antibiotics in the range of 40% to 80% with O_3_ treatment. In this process, the antibiotics are oxidized by O_3_ molecules or hydroxyl radicles after O_3_ decomposition [94]. Employing sand filter units alone does not effectively reduce the concentration of antibiotics, as the hydrophilic nature of most antibiotics means they remain suspended in the liquid phase. The exception is the removal of sulfamethoxazole with reduction rates in the range of 34% to 95% in sand filters [95,96]. They can be used in combination with other tertiary methods to improve performance. There is significant variation of antibiotic removal efficacy in WWTPs, indicating that the removal depends not only on the wastewater matrix but also on seasonal variations along with physicochemical properties of the antibiotics. Other factors such as degradation rates in the water, the acid dissociation constant, water solubility, and organic carbon water partition coefficients also assist in the removal from WWTPs [81]. The high removal rate of certain antibiotics such as quinolones is due to their adsorption into the sewage sludge, with a high sorption constant [97]. It has been proposed that complete removal of antibiotics is not possible in conventional WWTPs, and concerns remain about the discharged effluents being a threat to aquatic ecosystems and human health. Therefore, advanced technologies employing adsorption [98], advanced oxidation processes, such as Fenton-like oxidation [99], ozonation [100], sulfate radical-based oxidation [101], and ionizing radiation [102,103] are being recently used for the elimination of antibiotic residues.

#### 3.2.2. WWTPs and ARB Reduction

ARBs such as antibiotic-resistant *E. coli* and *Enterococci* are present in raw sewage and final treated effluent in WWTPs. These bacteria can display a range of antibiotic resistance phenotypes, and hence their removal in WWTP is of importance. The authors of [104] investigated both *E. coli* and *Enterococci* isolated from wastewater and found between 0.7–100% of the isolates in that study displayed at least one resistance phenotype. They also determined that bacterial species that are resistant to more than three antibiotics were about 25% in the influent and less than 20% in wastewater that was subjected to chlorination. A different study reported that the effluent of a WWTP consists of approximately 9.9% resistant *E. coli* and 0.2% total resistant coliforms. Based on previous reports, a two-log reduction of *E. coli* and *Enterococci* in WWTP is possible [105,106], including a 2 to 5 log reduction of resistant bacteria [107,108].

##### Primary Treatment Data/Literature

There is a diverse community of ARBs present in the influent of WWTPs, including untreated wastewater and hospital wastewater, providing evidence for the presence of clinically relevant ARBs. In particular, there is a lack of studies reporting the reduction of ARBs in primary units. Though selective pressure due to the presence of heavy metals, antibiotic residues, and other pollutants has the potential to assist in the amplification and spread of ARBs to the receiving environment, appropriate technology can reduce or eliminate them in the final effluent. Physical processes such as primary screening, grit removal and sedimentation remove the majority of the suspended solids, along with adhered bacteria. Primary screening does not remove/eliminate any suspended ARBs, as the screens are used for removing larger floating materials in the wastewater. Sedimentation also does not support significant removal of ARBs, as reported by [109,110]. However, a reduction in the range of 0–1 logs, as indicated by [111], has been reported. The removal of pathogens and ARBs is usually not high in primary treatment; therefore, it is important that downstream treatment units should efficiently remove ARB to avoid health risks.

##### Secondary Treatment Data/Literature

Teshome [112] has investigated activated sludge systems, showing they perform better as compared to septic tank systems for the removal of bacteria. The load of total and fecal coliforms, *Enterococci* species and *E. coli* in the raw influent was reported to be approximately 5.14 × 10^8^, 2.45 × 10^7^, 1.31 × 10^8^, and 1.17 × 10^7^ cfu/100 mL, respectively. Following activated sludge treatment, these numbers were reduced to 3.18 × 10^5^, 5.12 × 10^4^, 3.93 × 10^5^, and 2.75 × 10^4^ cfu/100 mL, with a log reduction of 3.21, 2.68, 2.52 and 2.62, respectively. This was comparable with the log reduction obtained after treatment with a septic tank system that ranged between 1.33 to 1.59 cfu/100 mL. The study also indicated that although there was a reduction of the bacterial population, there was an increase in the multidrug-resistant profiles of the isolates during wastewater treatment. It can be inferred that WWTP do provide a suitable environment for the proliferation of ARBs containing a high proportion of resistant bacteria. Berrios-Hernandez [113] inferred that the removal of fecal indicator organisms is not correlated to water quality parameters but is generally dependent upon the treatment technology used. In the case of *E. coli*, total suspended solids have been positively correlated with its removal, supporting the efficacy of primary and secondary treatment units for the removal of the microbial population. Activated sludge and novel aerobic granular sludge demonstrated somewhat similar removal efficiencies.

##### Tertiary Treatment Data/Literature

The relationship between chlorination and antibiotic resistance is complex. Disinfection in WWTPs includes the use of chlorine, UV irradiation and ozone. Chlorine is widely used in WWTPs due to its cost-effective nature and its beneficial oxidizing capacity for reducing the microbial population and pathogens in wastewater [114]. Chlorination used for tertiary treatment results in inactivation of bacterial cells and reduction of bacterial numbers, as measured by both culture and 16S rRNA [115]. Generally, chlorination, as currently used in tertiary treatment, induces the development of antibiotic resistance and co-selection of ARBs due to intermediate disinfection by-products (DBP), as reported by [116], or by inducing expression of multidrug efflux pumps. However, studies have reported that a fixed contact time (CT) value for a short duration of time combined with a high concentration of applied chlorine can decrease the abundance and reactivation of ARBs [117]. In contrast, longer CT combined with low chlorine concentrations may contribute to accelerated reactivation and regrowth of ARBs [118]. Other factors that contribute to the persistence of ARBs in WWTPs include a positive correlation with suspended solids and the chemical oxygen demand (COD) of untreated wastewater and a negative correlation with the temperature and dissolved oxygen (DO) of the effluent [119]. Several other factors, such as the type of antibiotic resistance, increase in ARBs due to the type of raw influent and environment of WWTPs, and decrease in antibiotic-sensitive bacteria in the bacterial population of the treated effluent, impacts the proportion of ARBs [120] and contributes to ARB load in WWTP. Antibiotic-resistant strains are frequently reported in the effluent, for example, *E. coli, C. perfringens,* fecal *enterococci* and *Sphingomonas* by [121], or ampicillin-resistant and tetracycline-resistant *E. coli* along with chloramphenicol- and cephalothin-resistant bacteria [122]. The authors of [115] identified *E. coli, Citrobacter* and *Enterococcus* for their survivability at 0.5 mg/L of free chlorine concentration. Bacterial cells that survive chlorination can re-grow and remain a concern for public health [123,124]. UV has been found to be beneficial for the reduction of ARBs [125] and their genes [126]. UV radiation acts directly on the DNA of microbes and does not react with ammonia or residual chlorine of treated wastewater. A recent study conducted by [1] reports reductions of 1–3 logs in facultative pathogenic bacteria after conventional wastewater treatment with UV. The major disadvantages of UV are that it is not a continuous process, as it provides time for photo-reactivation after the disinfection process. It also requires a high dose to eliminate ARBs, which is not practical in the applied field settings.

Ozonation has also been shown to impact the removal of ARBs by inactivation of bacterial populations (including antibiotic-resistant bacteria) due to the production of highly reactive radicles, though it is generally employed for the removal of organic micropollutants [127]. Additionally, ozonation followed by a filter passage can further lead to the reduction of total coliforms, *E. coli, Staphylococcus* and *Enterococcus* in the range of 0.8 to 1.1 log units when compared to flocculation filtration. *Enterobacteria*, *Staphylococcus*, *Enterococci*, and *P. aurigonosa* have demonstrated varying removal rates in the range of 60.2% to 98.9% at a concentration of 0.9 ± 0.1 g O_3_ DOC-1 (dissolved organic carbon), which is considered a regular dose applied for ozonation [128]. A log reduction in the range of 2.7–3.7 units has been reported with an ozone concentration that varies in the range of 5 mg/L to 10 mg/L for *E. coli* present in WWTP [129]. A high concentration of ozone in the range of 25–30 mg/L can reduce coliforms by 4.4 log units, *Enterococci* by approximately 3.6 log units and *Clostridia* in the range of 1.2–1.7 log units [130]. The resistances of the target microbes, the degree of damage due to ozonation, and the quality of wastewater are some of the factors that assist in the further reduction of ARBs in ozonation treatment. Combinations of different units are effective for reduction in abundance for *Pseudomonas, Enterococcus* and *E. coli* [131,132]. The conventional treatment of wastewater that includes removal of solids, sediments, grease, sludge [133], biological removal of nutrients such as nitrates and phosphates [134] along with tertiary treatment for disinfection [135], use of CWs [136], advanced oxidation process (AOP) [137] and hybrid units are efficient for the elimination of a large number of ARBs [138]. There remain many knowledge gaps on the regrowth and reactivation of ARBs after conventional treatment.

#### 3.2.3. WWTPs and ARG Reduction

WWTPs provide suitable temperature, pH, nutrients, and environment for the growth and proliferation of bacteria, transfer of ARGs, and spread of both to the receiving water bodies or soil [138]. Generally, it has been found that the concentration of ARGs in influent and effluent vary according to treatment technology. Bacteria carrying multidrug resistance and ESBL genes such as *E. coli, Enterobacter* species, and *Salmonella* species can be detected in treated effluent. The ARGs code for resistances associated with health risk threats and are especially concerning when associated with pathogens. The impact of WWTP-associated ARGs on environmental bacteria is unknown but is of potential concern.

##### Primary Treatment Data/Literature

As reported by [139], a negligible amount of ARGs are removed after the reduction of suspended solids in primary treatment. It may range between 0.09 and 0.55 orders of magnitude [140] and also, as reported by [141], in the range of 0.17 to 0.5 logs. The sedimentable solids generated that settle in primary settling tanks contain ARGs, which may enter the environment if the solids are used as fertilizer without any sludge treatment, such as the application of lime. ARGs and their host bacteria may be recirculated along with the sludge (used for seeding purposes) into the WWTP system, further increasing ARG abundance in the treatment plant [142].

##### Secondary Treatment Data/Literature

Reports of ARG removal efficacy after secondary treatment are mixed and depend on a complex interaction of factors. Some reports have confirmed 1–2 log reductions of ARGs [143], while others report 1.3–6.1 log reduction of ARGs [107,119,144]. Higher rates of reduction have been observed due to the removal of suspended solids (in primary treatment) and activated sludge processes (in secondary treatment) [139]. Stiborova [145] concluded that dewatered activated sludge promotes the development of multidrug resistance due to adsorption on its solid surface, acting as a reservoir of ARGs when used as fertilizer in agricultural soils.

##### Tertiary Treatment Data/Literature

Tertiary treatment plays a significant role in lowering the abundance of ARGs. A recent study by [146] reported the reduction of total 16s rRNA, ARGs and *int*1 by 0.1 ± 0.4 log 10 copies/mL, *tet*A by 0.9 ± 0.6 log 10 copies/mL, *erm*F by approximately 0.1 ± 0.3 log 10 copies/mL, blaTEM by 0.4 ± 0.5 log 10 copies/mL, and *int*1 by 0.5 ± 0.4 log 10 copies/mL. They reported the same type of variation in another treatment plant, indicating a reduction in the abundance of ARGs. The final effluent of 2 WWTP released 3.3 ± 1.5 log 10 copies/mL and 3.4 ± 1.4 log 10 copies/mL of total ARGs and *int*1. Therefore, their results demonstrate that the conventional WWTPs using chlorination can be confidently used to avoid the proliferation of ARGs during treatment. ARGs are naturally carried within bacterial host cells, either chromosomally or on plasmids. Upon lysis of the host bacteria, ARGs are released into the environment where they persist until they are taken up by a new host or are degraded and lose their biological activity. However, even when chlorine and other disinfectants kill bacteria, they do not necessarily inactivate or degrade the ARGs [147]. While effective for reducing viability in bacteria, cells that survive chlorine treatment appear to be primed to uptake foreign genes, including ARGs released from recently killed bacteria [148]. The hydroxyl radical and reactive chlorine species formed result in DNA damage and may also alter the genetic characteristics [149].

High CT favors the reduction of ARGs when they are present at high concentrations but is less effective when ARG targets are present in lower concentrations. For example, some evidence suggests that certain ARGs may be more easily inactivated by high CT (>80 mg Cl_2_ min L^−1^) as compared to low CT (<10 mg Cl_2_ min L^−1^) than others [150], though it is not clear yet what other factors may impact these results. ARG removal can be observed beyond the breakpoint of chlorination [151] with an added advantage of oxidizing organic matter and removing color and odor from wastewater. In contrast, a study by [122] revealed that 80% and 40% of tetracycline- and erythromycin-resistant genes, respectively, could not be removed even after chlorination with a CT value of 15 mg Cl_2_/min/L (average range is 15–30 mg Cl_2_/min/L with different CT, which is quite efficient for reduction of ARGs in wastewater). The residual chlorine in the effluent reactivates the development of ARGs [152]. It also promotes HGT and alters the permeability of bacterial cells [153]. Zhang [117] reported the maximum removal of ARGs to be up to a 1.49 log 10 reduction, which was obtained with a free chlorine dose of about 30 mg/L and 30 min contact time.

Proper dosing of chlorine with CT (and focusing on by-products obtained) coupled with natural treatment systems (such as CWs, where nutrients and various pollutants present in wastewater are broken down/degraded naturally and taken up by plants and bacterial communities, removing them from wastewater) prevents the growth of pathogenic bacteria and is a useful option for effluent to be discharged to the environmental matrix [154,155]. Moreover, while considering the role of tertiary treatment in the removal of ARGs, their dose of application (O_3_, UV and chlorine), contact time with the wastewater and wastewater characteristics play a significant role [4,65,156]. Additionally, investigations have reported that there are times when O_3_ is relatively less effective than the other two disinfection methods, as O_3_ gets generally consumed by the organic matter in the effluent [157]. When coupled with UV treatment, chlorination is reported to be modestly effective for the removal of ARG, integrons that are associated with horizontal gene transfer, and 16s rRNA genes that serve as a marker for bacterial cells [158]. Despite the fact that ARGs can persist following tertiary treatments, disinfection methods have been shown to be successful for the reduction of ARGs. There are multiple factors that allow for the reduction of ARGs in tertiary treatment. The type and process used for wastewater treatment, location specifics, seasonal influence, the concentration of ARGs already present in the influent, the concentration and type of antibiotics and other contaminants in the influent, and survival factors of the host bacterial taxa all impact the efficacy of tertiary treatment for the reduction of ARGs. Other factors that impact ARG removal are initial ARG concentration and water quality, especially the concentration of ammonia nitrogen in wastewater (>15 mg/L), with a reduction of approximately 1.49 log10 ARG [146]. Furthermore, appropriate pre-treatment of sludge with free ammonia is able to enhance anaerobic digestion, as reported by [159]. Figure 2 consolidates the efficiency of removing antibiotics and resistant genes by conventional and tertiary treatment technologies, including CWs.

### 3.3. Performance of CW for Elimination of Antibiotics, Antibiotic-Resistant Bacteria and Genes

There is an emerging awareness of the value provided by biological treatment processes employing CWs [160,161]. CWs have a long history of being utilized for pollutant removal, and research has shown them to be capable of significant reductions of organic pollutants, nutrients [162], heavy metals [163,164], antibiotics, and emergent contaminants such as ARG/ARB [32,165]. They can be configured in many ways and tailored to meet individual needs and specifications. Appendix A consolidates the comparative techno-economic assessment of CWs over other technologies.

#### 3.3.1. Efficiency of CWs for Removal of Antibiotics

Antibiotic drug residues are released to the environment through WWTP and can affect the efficiency of the microbial communities essential to CW function. Antibiotics inhibit some types of bacterial growth [166], impact denitrifying bacteria [167,168] and potentially alter the denitrification rate [169]. Remediation of wastewater containing a high percentage of nitrate utilizes biological denitrification, where CW plays a significant role [170]. Other important CW functions that rely on microbial processes are the removal of organic carbon (the soluble labile forms present in primary treated domestic wastewater), the removal of pollutants such as heavy metals, and the removal of total nitrogen [171]. The reduction of sulfate also plays a significant role in metal removal, and hence, sulfide oxidation is an important process in CWs. Pollutant removal and microbial activity in CWs are closely associated with nitrogen, sulphur and carbon cycling [171]. Considering the introduction of antibiotics to CWs, for example, the presence of ciprofloxacin in wastewater influent leads to an increase in resistance for other classes of antibiotics such as penicillins, tetracyclines, sulfonamides, and cephalosporins, including ciprofloxacin. Antibiotics have the potential of increasing the antibiotic resistance profile of others present [172]. Studies by [173] evaluated the high concentration of erythromycin and sulfamethoxazole in the influent of CW. They reported microbial degradation plays a significant role in the reduction of these drugs (13–99%) from wastewater.

A growing body of work exists examining the efficiency of various CW systems to remove individual antibiotic drug targets. One theme that emerges from the work done to date is that the diversity of CW configurations, process parameters, and site-specific differences can impact drug removal efficiency. Li [174] evaluated the removal of tetracycline in an integrated VFCW with three days of hydraulic retention time (HRT) and continuous flow of domestic wastewater and reported up to 75% removal efficiency, similar to earlier reports by [83]. Tetracycline and quinolones are less soluble and possess a greater tendency to get adsorbed to substrates, where they undergo photodegradation due to solar influx. VFCWs have also been reported to reduce sulfamethazine, sulfadiazine and sulfamethoxazole in wastewater, with removal efficiencies in the range of 26.42–84.05% [175]. Nitrification was slightly enhanced due to the presence of sulfamethoxazole. It was confirmed that both aerobic and anaerobic microbial degradation assists in the removal of sulfonamides from VFCWs. Sulfonamides did not influence total phosphorus (TP), total organic carbon (TOC), or ammonia-nitrogen (NH_4_^+^-N) removal but increased the diversity and altered the community structure of bacteria. It is indicated that the antibiotics induce selective pressure on the bacterial population, rendering them resistant to a wide range of antibiotics.

Vertical subsurface flow CWs (VSSFCW) are basically operated by providing intermittent feeding (of wastewater) from the top that drains to the bottom of a filter bed. The intermittent flow allows the process of nitrification in the unsaturated zone, promoting microbial degradation of antibiotics in a way that is superior as compared to HSSF-CW, which remains saturated all through [176]. Hence, these unsaturated phases between pulses allow an enhanced nitrification capacity for the removal of NH_4_-N and organic matter [177]. The denitrification process is generally limited due to the lack of an anaerobic area that is required for the growth of denitrifying bacteria. The microbial communities in the VSSFCW allow complete nitrogen transformation through nitrification/denitrification, as the microbes are metabolically active and play a significant role accordingly. Reports by [178] highlighted that microbial degradation was responsible for the up to 94% reduction observed for tetracycline and enrofloxacin. The partial saturation in the lower zone of the media (that is comprised of gravel or other media) promoted redox gradients throughout the filter beds [179]. Benefits of VSSFCWs include higher removal of pollutants, especially emergent contaminants such as antibiotics (sulfamethoxazole, erythromycin, trimethoprim, etc.) and ARGs [180] compared to standard WWTPs. Liu [181] have evaluated 106 CW and measured their removal of 39 classes of antibiotics as well as antibiotic resistance genes. It was found that VSSFCWs were the most efficient of all the configurations and showed an average removal of 70% of the antibiotics. The VSSF had different ranges of operating parameters, including temperature, solar flux, type of influent, and concentration of antibiotic residues. Physical characteristics such as water solubility and octanol-water distribution coefficient (Kow) of the antibiotic compounds affect its removal from CW. A study [182] working in a VFCW system reported the removal of ciprofloxacin in the range of 83–93% and sulfamethazine in the range of 56–69%. Liu [181] demonstrated an 81% reduction in macrolide concentration in a VSSF-CW. The mechanism was thought to be a combination of high Kow values, less water solubility and high adsorption of the drug onto the substrate and plant roots. These results highlight the complexity of antibiotics in the environment and the urgent need to identify which measurements are most relevant for assessing risk to human, animal, and environmental health.

Sochachi [183] evaluated removal of sulfamethoxazole in the range of 52.8–91.2%, where the VFCWs were planted with reed canary grass ‘*Picta*’ (*Phalaris arundinacea* L. var. *picta* L) that was used for the treatment of urban wastewater. It was observed that the presence of antibiotics or pharmaceuticals and their transformation products had a direct impact on the floating plants and resulted in deterioration and decay, thereby showing phytotoxic activity. It was also concluded that despite high concentrations of antibiotics/pharmaceutical, the emergent macrophytic species, namely *Gaint Miscanthus*, in subsurface flow constructed wetlands (SSFCW) demonstrated no alterations in their physical appearance and suggested the use of hybrid systems for SSFCW and SFCW. Integrated flow constructed wetlands (IFCW) use a combination of different CW. IFCWs demonstrate better efficiency compared to a single unit of CW mainly for the removal of nutrients and other pollutants [184]. Choi [185] investigated *Pharagmites australis* and *Triarrhena sacchariflora*, which were found to efficiently remove sulfadiazine at rates up to 81.86%, sulfamethoxazole at rates up to 85%, sulfamethazine at rates up to 49.43% and trimethoprim at rates up to 2.32% with a retention time of 48 h. *Chrysopogon zizanioides* has also been used to remove antibiotics, especially tetracycline and ciprofloxacin, from secondary wastewater effluent in the presence of a high concentration of nutrients (specifically N and P). The removal percentage varied between 89–100% for tetracycline and 60–94% for ciprofloxacin, and there was an inverse relationship between nutrient concentration and antibiotic removal [186]. Lab-based studies have reported the significance of plant species associated with microbial communities in efficiently removing antibiotics. Kurade [187], in a batch study, selected *Ipomea aquatic* based on its potential for remediation of 100% sulfamethoxazole in 30 h. It was observed that the primary mechanism of sulfamethoxazole elimination was biodegradation, which accounted for 82% of the observed reduction. Results demonstrated that *Ipomea* converted sulfamethoxazole into simple compounds like 4-aminophenol as its end product. Plant roots assisted in bioadsorption but accounted for only 0.77% of the drug removal, and leaves bioaccumulated up to 16.94% of the sulfamethoxazole. An 8% increase in chlorophyll and a 9% decrease in carotenoids was observed after 48 h of exposure to sulfamethoxazole, but this did not induce any toxic effect on the photosynthetic activity of *Ipomea*. It has a defense mechanism against toxicants and abiotic stress which is activated during the initial exposure to sulfamethoxazole. In a different study, the efficiency of CW planted with *Juncus acutus* was utilized for the removal of ciprofloxacin at rates of up to 93.9%, but no significant removal was observed for sulfamethoxazole [188]. *Juncus acutus* is a halophyte tolerant to different stressors, specifically organic xenobiotics, heavy metals, endocrine-disrupting chemicals, pharmaceuticals and personal care products. This makes it a particularly good candidate for CW. Only 2% of plant species (of the known plant species) are halophytes that demonstrate good phytoremedial potential [189,190,191]. Along with metal accumulation, phytostabilization has also been reported as another remediation mechanism in the root structure of *J. acutus* [188].

#### 3.3.2. Efficiency of CWs for Removal of ARBs and ARGs

In addition to the ability to reduce the concentration of antibiotic drugs in wastewater systems, CWs have also been evaluated as a means to mitigate ARBs and ARGs. Figure 3 highlights, in brief, the protocols used for antibiotic resistance, factors associated to encourage this review study, along with the benefits and use of CWs. In this regard, the large body of work evaluating the efficacy of CWs to reduce fecal bacteria, including fecal pathogens, helps to inform our understanding of the potential of CWs for remediating antibiotic-resistant bacteria of both fecal and environmental origin. Previous studies have already reported the removal of pathogenic and fecal indicator bacteria in a variety of CW systems [192,193]. As with WWTP, positive correlations have been reported for CW between microbial parameters and NH_4_, total nitrogen (TN), and a negative correlative between *E. coli* and total suspended solids (TSS), TP, COD and biochemical oxygen demand (BOD) [194]. Lamori [195] highlights that the *E. coli* concentration decreased up to 50% in the effluent of a CW as compared to the influent. In the study, they compared the presence of *E. coli*, Enterococci, Bacteriodales (HF183) and ARG (*erm*(F) and *int*1) in the influent and the treated effluent. It was demonstrated that *erm*(F) and *int* were reduced by 13% and 67.2%. The study also found that there was no correlation between turbidity and ARG, though sediments act as a reservoir of ARGs. The growth of *E. coli* was affected by water temperature, sedimentation rate, pollution source and overall bacterial growth. At higher temperatures (especially during the summers), the reduction of *E. coli* (that was present in 100% of the water samples) ranged between 31 and 70% and *Enterococci* between 70 and 99%. The temperature had an impact on microbial composition in treatment plants that affected bacterial removal rates. It was indicated that the microbes grow and survive at a pH range of 5.5–7.5 and are pH-sensitive [196]. High pH was associated with increased bacterial/pathogen removal in CWs. The removal of pathogens also depends upon the physicochemical factors of wastewater [32] and the source of treated effluent. It is indicated that the overall bacteria decrease but are still abundant in the final effluent of a conventional treatment plant. CWs further reduce the concentration of bacteria due to vegetation, temperature, sunlight, high retention time, etc. Furthermore, as reported by [197], during winters, the removal rate of ARGs is considerably high as compared to summers, most probably due to low temperature.

Hien [198] investigated the removal of five tetracycline-resistant genes (namely *tet* C, M, O, Q and W) and reported their complete removal by a CW, with 1 day of retention time, using *Pharagmites australis*. The CW demonstrated a reduction of up to 3 logs for ARGs, which increased with an HRT of 2–3 days. A comparative work recently reported by [199] investigated the abundance of sulfonamide- and tetracycline-resistant genes in the influent and effluent of WWTPs with and without a CW. One WWTP using a conventional activated sludge (AS) system and a second WWTP coupling a CW with AS were evaluated for ARG removal potential. The CW was found to be more efficient for reducing ARG as compared to AS. Primarily, investigations have revealed that there exists a positive correlation between antibiotic concentration present in the influent and the abundance of resistant genes. One study reported a high concentration of ARG attributed to the accumulation of antibiotics in the bottom layer of the CW. Specifically, the *sulI* and tetracycline genes were found to be persistent due to the selective pressure by antimicrobial agents [200]. Multiple CW designs have been compared for their relative effectiveness in reducing ARB/ARG over time. SSFCWs were superior to SFCW for the elimination/removal of selected ARG targets [201]. In another study, however, they act as reservoirs for specific ARGs [197]. There was a strong positive correlation between the *int1* gene and ARGs, suggesting that MGE affects the dissemination of ARGs in CW. The fate of ARGs *sulI, sulII* (sulfadiazine), *tet*M, *tet*O, *tet*Q, *tet*W (from tetracycline) and 16S rDNA genes was evaluated in a HSSF-CW. It was found that the HSSF-CW performed better than the conventional treatment plant with aerated filters or UV disinfection. The reduction of ARG was in the order of 1–3 log units after treatment [139]. Miller [202] has reported the efficiency of CW for the removal of ARGs, especially considering the HLR and HRT. In VSSF-CW, high infiltration and HLR lead to the filtering out of bacteria and help bind extracellular DNA onto the soil particles. A long duration of HRT allows the microbes to degrade the antibiotics, allowing ample time for the process to occur in the CW bed. Microbial degradation also depends on the seasons, where high temperatures allow better survival of bacterial populations. The VSSF CW showed an increase in ARG removal when substrates like zeolite were used, increasing the relative surface area and Si-OH structures for chemical sorption along with degradation capacity of the microbes [203]. Table 2 consolidates a few recent studies highlighting CWs for the efficient reduction of ARB/ARG from various wastewater sources.

### 3.4. Underlying Factors and Mechanisms Involved in the Reduction of Antibiotics, ARB and ARG in CW

The primary mechanisms of the removal of pollutants in CW are sedimentation, sunlight/UV radiation exposure, inactivation/death associated with unfavorable water chemistry, oxidation, biofilm interaction [210], predation, antibiotic or biocides secretion and natural die-off [211]. One technique that has been used to increase the capacity for ARG removal from CWs is to recirculate the effluent into the CW to increase the binding of extracellular DNA into the substrate [212]. Studies have reported that adsorption onto the substrate and biodegradation by the microbial population are two significant mechanisms for the removal of ARGs in CW [55,213,214]. Economics are a factor when deciding on CW substrates. Some substrates are expensive (like zeolite, oyster shell, etc.) but may be used in certain ratios to increase the microbial attachment or chemical sorption or for removal of a particular pollutant of interest. Simultaneously, redox reactions also assist in the degradation of organic substances, as reported by [215], but studies relating to varying redox conditions and ARG removal have not been reported and need to be carried out.

### 3.5. Role of Plants in CWs

Plant species, whether mono or mixed culture, are effective for the removal of antibiotic residues, ARG and other pollutants from wastewater, and accordingly, they can be selected on the basis of removal of a particular pollutant [203]. Vegetation does not directly remove ARG, but it provides oxygen to microbes, provides a surface to biofilm development and filters the solid particles. A planted CW has shown ARB removal of up to 81–93% as compared to unplanted constructed wetlands (42–74%) [197]. Planted CWs have root structures that harbor populations of microbes that degrade or transform antibiotic drugs, antibiotic-resistant bacteria, and ARGs. Microorganisms are the drivers of antibiotic drug degradation, utilizing it as a carbon source. Investigations have confirmed that the root exudate helps in the degradation of pollutants and adapts the plant to the toxic effects of antibiotics [31]. The presence of *Phragmites australis* in CW has shown encouraging removal of ARGs and 16sRNA levels compared to other plants. Other aquatic plants such as *Cyperus alternifolius, Pontederia cordata* and *Myriophyllum verticillatum* can also contribute to the removal of ARGs [36]. A recent study on *Pharagmites australis* shows a tolerance mechanism and biological response to ofloxacin and sulfamethoxazole residues at even a simulated condition of 1 mg/L. It was found that the growth of plants was not inhibited, and there was an improvement in root and photosynthetic activity. Low doses of antibiotics increased the transpiration and photosynthetic rate by stimulating the opening of stomata, which assisted the transportation of water and antibiotics and improved the efficiency of light utilization by acting upon the reaction center of the photosynthetic system. Individually, ofloxacin reduced the accumulation of reactive oxygen species in the plant. Both antibiotics accumulated prominently in the roots, then leaves, and finally the stem. Results demonstrated that *Pharagmites* is efficient in the removal of antibiotics from wastewater [214]. Studies by [216] reported that the bacterial composition in biofilms was determined by the type of substrate. The water flow increased the bacterial diversity in biofilms, and it regulated the distribution, diversity and biomass of epiphytic biofilms in CW with submerged macrophytes. Actinobacteria, Proteobacteria, Bacteroidetes, Firmicutes and Planctomycetes were found in the biofilm, whereas Proteobacteria, Bacteroidetes, Euryarchaeota, Thaumarchaeta were found in the sediments in different percentages. Appendix A demonstrates the role of macrophytes and the mechanism for the reduction of pollutants in CWs.

## 4. Conclusions and Implications

The presence of antibiotic resistance in wastewater effluents has stimulated recent studies for understanding the abundance, occurrence, fate and potential control in WWTP. CWs are another option. CWs are quite efficient for the reduction of general water pollutants, nutrients, and emerging contaminants, including antibiotic drugs, ARBs/ARGs, and they have demonstrated encouraging results in recent studies. CWs are considered to be a flexible treatment technology with the potential to reduce a wide variety of pollutants. CWs can be used as decentralized systems where each individual housing unit can have its own pretreatment unit as a septic tank/sedimentation tank, allowing flow to the CW and then the discharge of treated water. They can also be integrated with secondary and tertiary treatment in conventional wastewater treatment plants as a viable option for the reduction of the microbial load. The treated effluent can be used for gardening purposes, agriculture, artificial groundwater recharge, as direct discharge to water bodies and in industrial usages, like washing, cooling, etc. CWs find their application particularly in developing/low-income countries, where there are failures for treating wastewater due to lack of treatment plants, tertiary units, electricity and manpower cost. CWs reduce the cost of conventional treatment plants and work on a non-mechanized system. Other factors include local availability, simplified operation, reduced construction cost, ability to maintain at an individual level, enriched aesthetics to the environment, and environmental restoration. Parks and gardens can be designed according to space availability for public awareness and acceptance. According to use, multistage or hybrid CWs can be designed that further increase the performance efficiency of the system. CWs treat wastewater and make it fit for reuse purposes, providing an alternative for large volumes of effluent from wastewater treatment plants. This would otherwise be difficult to manage and pose a potential threat to the receiving environment. Hence, the use of CWs as a flexible treatment technology is recommended, with the potential for reducing antibiotic resistance. Multistage and hybrid CWs are effective for treating raw urban wastewater. The technology, if adopted, will help in meeting the national action plans in developing and low-income countries, with options to use the treated effluent for multiple purposes. Investigations are further required in the relevant direction focusing on (i) the effect of antibiotic usage and its relationship with wastewater generated from the domestic and industrial sector, (ii) the impact of residual antibiotics in the influent on the selection pressure of ARG to promote resistance, (iii) the concentration of commonly used antibiotics in the influent/effluent of WWTP that is discharged into the water bodies/agricultural land, (iv) the detection of commonly available ARGs in terms of type and concentration by the latest techniques like qPCR, (v) the lacking correlation of different water quality factors and mechanisms, which needs to be explored further, (vi) the reactivation of ARG in the effluent after chlorination or treated effluent of CWs in the water or soil matrix and the overall impact of ARG disseminated, and (vii) identifying the mechanisms underlying the removal of ARG/ARB removal in CW that is essential for designing highly effective CWs for removal of antibiotic drugs, antibiotic-resistant bacteria, and ARG.

## Figures and Tables

**Figure 1 antibiotics-11-00114-f001:**
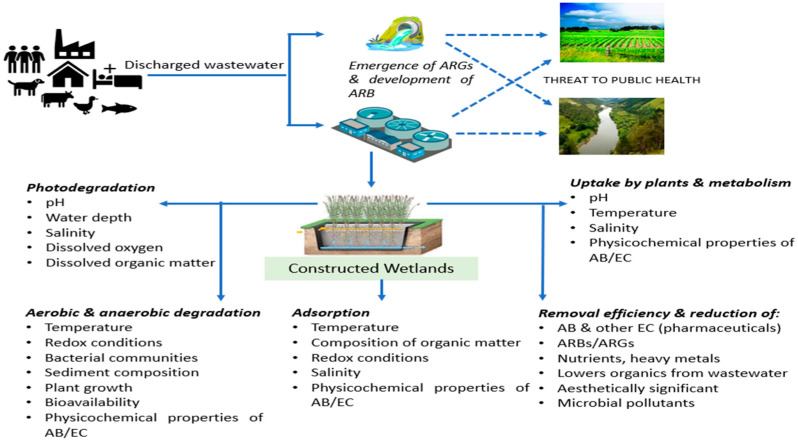
Source, development, pathway and potential threats of ARB/ARGs on the environment, and application of constructed wetlands through various mechanisms for the reduction of the same (Note: ARGs—antibiotic resistant genes; ARB—antibiotic resistant bacteria; AB—antibiotics, EC—emerging contaminants).

**Figure 2 antibiotics-11-00114-f002:**
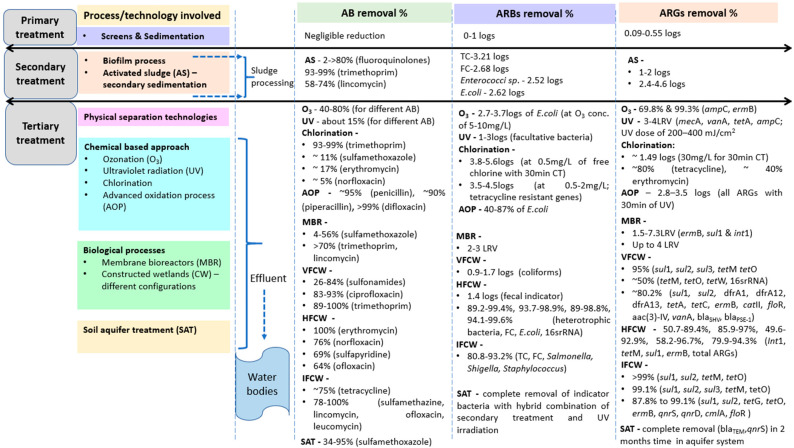
Removal of antibiotics, antibiotic-resistant bacteria and genes by conventional and tertiary treatment technology, including constructed wetlands (Note: VFCW-vertical flow CW, HFCW-horizontal flow CW, IFCW-integrated flow CW).

**Figure 3 antibiotics-11-00114-f003:**
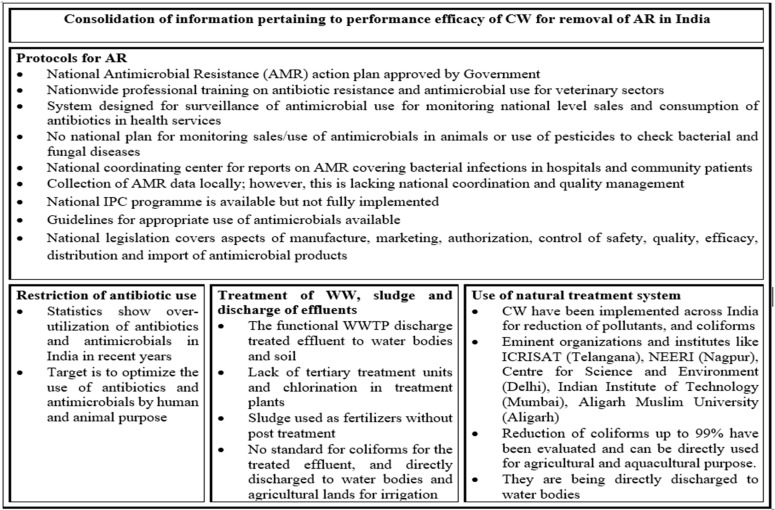
Flow diagram representing, in brief, the protocols for antimicrobial resistance and benefits of using CWs in various organizations/institutes in India.

**Table 1 antibiotics-11-00114-t001:** Description of the question of concern for the present review study related to antibiotic resistance in wastewater and the use of constructed wetlands in reducing the loads from treated effluents.

Q1. Protocols for AR	Q2. Treatment of WW, Sludge and Discharge of Effluents	Q3. Use of Natural Treatment System
What are the protocols/interventions implemented in India for the control of AR in the environment?	What are the technologies that assist in reducing the load of pollutants from wastewater? Are they able to reduce antibiotics, ARB and ARG load?	Are CWs efficient enough for the control of antibiotic resistance from the effluent of treatment plants? How effective is it, and can they be implemented at a larger scale?
** *Source:* **		
National action plan on antimicrobial resistance (2017)	WW and sludge generated through urban sectors in both liquid and solid state or a mixture of both Main pollutants: antibiotic drugs, ARG/ARB	Any water matrix contaminated through domestic and industrial wastewater, e.g., ponds, lakes, rivers, canals Main pollutant: antibiotic drugs, ARG/ARB
** *Primary objective:* **		
Effective understanding of AMR through trainings, awareness and practice of health care in the community Optimize the use of antibiotics/antimicrobials in veterinary and human use through awareness, stewardship, and effective regulatory mechanisms for markets Promote innovations and research on AMR and collaborate with international organizations	Treatment of WW: primary, secondary and tertiary treatment through conventional methods in WWTP, as well as aerobic or anaerobic digestion Post-treatment of sludge: drying [50], vermicomposting [51,52,53], liming [54]	Use of NTS for efficient reduction of coliforms and AR from effluents of conventional treatment technologies [55] Various designs and configurations, use of substrates and plant species assist in better removal [56]
** *Pathways of exposure/disposal:* **		
Veterinary and human uses, wastewater from domestic and industrial discharges, agricultural runoff	Treated effluents can be polished if CWs are coupled as tertiary units in WWTP [57] CW further assist in reducing the pollutant load of pathogens, bacteria, coliform, ARG/ARB [58] and other emergent pollutants like pesticides, antibiotics [59,60]	Disposal of treated effluent for multifarious uses such as reuse in toilets, aquaculture, gardening and agricultural purposes [61], direct discharge to water bodies, recharge to groundwater [62]
***AR in the environment:*** Wastewater treatment plants, sludge, effluent discharges from pharmaceutical industries, domestic and other industrial wastewater, hospital discharges, agricultural runoff, poultry and veterinary wastes, swine and feedlot wastewater, soil and sediments		
***Types of recent studies in AR:*** Clinical studies: case studies on humans, animals, plants Environmental studies: laboratory and field-based studies on water, soil, agricultural farms, poultry, review of previous study and comparative study involving effects of anthropogenic activities		

Note: AMR—antimicrobial resistance, AR—antibiotic resistance, ARB/ARG—antibiotic-resistant bacteria/antibiotic-resistant genes, CW—constructed wetlands, WWTP—wastewater treatment plants, NTS—natural treatment system.

**Table 2 antibiotics-11-00114-t002:** Removal efficiency of various CWs for reduction of ARB and resistant genes.

Type of CW	Source/Type of Wastewater	Antibiotic-Resistant Bacteria/Antibiotic-Resistant Genes	Removal Efficiency	References
Hybrid/integrated flow CW	Mixed wastewater from restaurant, hostel & brewery	*E. coli* and thermotolerant coliforms	99.5%	[204]
	Dairy wastewater	Total coliforms, fecal coliforms, fecal streptococci, and *E. coli*	99.93–99.99%	[205]
	Raw domestic wastewater	*sul*2, *erm*B, *cml*A and *flo*R	87.8% to 99.1%	[206]
	Domestic wastewater	Enterococci, *HF183, intI1*, and *ermF*	84.0%, 66.6%, 67.2% & 13.1%	[195]
	Raw landfill leachate	Sulfonamide resistance (*sul*1, *sul*2, and dfrA), aminoglycoside resistance (aac6), tetracycline resistance (*tet*O), quinolone resistance (*qnr*A), and *intl*1, beta-lactam resistance (bla_NDM1_, bla_KPC_, and bla_CTX_) and macrolide-lincosamide resistance (*erm*B)	>90%	[32]
	Rural domestic wastewater	*sul*1, *sul*2, *tet*M, and *tet*O	>99%	[36]
Vertical flow CW	Urban wastewater	*sul*1, *sul*2, *qnrS*, bla_TEM_, *erm*B	46–97%, 33–97%, 9–99%, 18–97% & 11–98%	[41]
	Swine wastewater	*sul*1, *sul*2, *sul*3, tetM and *tet*O	95.1%	[207]
	Swine wastewater	Tetracycline-resistance genes (*tet*) and integrase gene of Class 1 integrons	33.2 to 99.1%,	[208]
	Raw domestic wastewater	Sulfonamide resistance genes (*sul1, sul2* and *sul3*), four tetracycline resistance genes (*tet*G, *tet*M, *tet*O and *tet*X), two macrolide resistance genes (*erm*B and *ermC*), two chloramphenicol resistance genes (*cml*A and *flo*R) and 16S rRNA (bacteria)	63.9 and 84.0%	[203]
	Swine wastewater	Tetracycline resistance (*tet*) genes such as *tet*M, *tet*O, and *tet*W	Reduced by 50%	[209]
Horizontal flow CW	Raw domestic wastewater	*int*1, *tet*M, *sul*1, *erm*B and total ARGs	50.7–89.4%, 85.9–97%, 49.6–92.9%, 58.2–96.7% & 79.9–94.3%	[33]
	Hospital wastewater	Tetracycline-, erythromycin- and ampicillin-resistant and higher multidrug-resistant bacteria	80.8% to 93.2%	[26]

## Data Availability

Not applicable.

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
