# Peer review of "Performance Efficiency of Conventional Treatment Plants and Constructed Wetlands towards Reduction of Antibiotic Resistance"

_antibiotics, 2022, doi:10.3390/antibiotics11010114_

Round 1

Reviewer 1 Report

This review is a good, well-structured and reasoned article on a problem of great global importance. To counteract the phenomenon of resistance it is important to know well all the mechanisms and ways to reduce the presence of ARB and ARGs.

Here is just a small comment that could improve your manuscript: - in paragraph 2.1 "General approach" it would be appropriate to indicate the review period taken into consideration, the keywords searched, the inclusion and exclusion criteria. 

Reviewer 2 Report

The some findings of the study are already published in several journals, in my opinion there is no new points that has been highlighted in this manuscript. I agreed authors have reviewed varied number of articles with remarkable methodology. It would be nice if some novel aspect can be introduce through this manuscript.

Reviewer 3 Report

The manuscript by Moushumi Hazra and Lisa Durso titled: "Performance Efficiency of Conventional Treatment Plants and Constructed Wetlands towards Reduction of Antibiotic-Resistance" addresses the very important topic of presence and means of elimination of antibiotics, ARBs, and ARGs from wastewater in both conventional treatment plants and constructed wetlands. The review presented contains a comprehensive analysis of the current literature. The manuscript has been prepared carefully and concisely, maintaining a logical sequence of information presented. The article also contains numerous summary tables and figures to help the reader summarize the presented knowledge. 
The paper requires minor corrections:
- In Lines 34-35 the phrase "microbial flora" was used - please change the phrase flora to "communities" or other;
- Low quality of Figure A1 in supplementary materials.

Reviewer 4 Report

I appreciate generally this review as to be on a very important and interesting topic. The review is written in a very clear and logic way (through steps of the antibiotic reduction, ARB reduction and ARG reduction) and considering single factors influencing the reduction of ATB, ARB and ARG with a high amount of used literature sources. From this reason I recommend to publish this review only with few minor, mainly formal revisions (see below).

Minor revisions:

1) The constructed wetlands presented in Figure 1 (page 2) are in this figure named as "green innovative technology". I suppose it can misleading to the reader as this term is not used or more discussed in the article itself. I recommend to name the constructed wetladns in this figure as constructed wetlands and then in the text possibly to add some other examples of ranking constructed wetlands into "green innovative technology" or to mention it in the caption of Figure 1.

2) L. 150-152: What does the following sentence mean ? "There is an increasingly robust literature on CWs and its potential technology that assist in removal of antibiotic resistance especially during 2019 and 2021." Does it mean that the assitance of CWs for removal of antibiotic resistnace has increased in 2019 or 2021 ? Do you expect the same trend to continue ?

3) In line 552 in one sentence it is named the grass Phalaris arundinacea. But in the following sentence the name for one of its cultivars Picta is used. It has to be explained that Picta is a cultivar of Phalaris arundinacea, which is not understable from the text.

4) I recommend to present Figure 3 (p. 14) as a table, as it is a table and listing this as a table will improve the graphical quality.

5) Please correct (l. 650-651) "Long duration of HRT allows the microbes to DEGRADATE the antibiotics which also dependS on the seasons, during which the high temperatures allows...".

6) In Table 2 the (p. 15) name of the third column is "Antibiotic resistant bacteria/Resistant Strain". I recommend to replace "Resistant Strain" with "Antibiotic resistance genes" as later ARGs are those which mention in Table 2.
